# Assessing an Outdoor Office Work Intervention: Exploring the Relevance of Measuring Frequency, Perceived Stress, Quality of Life and Connectedness to Nature

**DOI:** 10.3390/healthcare13141677

**Published:** 2025-07-11

**Authors:** Dorthe Djernis, Charlotte Petersson Troije, Victoria Linn Lygum, Peter Bentsen, Sidse Grangaard, Yun Ladegaard, Helle Haahr Nielsen, Katia Dupret, Christian Gaden Jensen

**Affiliations:** 1The Foundation for Mental Health, Høffdingsvej 22, Valby, 2500 Copenhagen, Denmark; cgj@fondenmentalsundhed.dk; 2Department of Urban Studies, The Faculty of Culture and Society, Malmö University, Nordenskiöldsgatan 1, 20506 Malmö, Sweden; charlotte.petersson@mau.se; 3Division of Sociology, School of Health, Care and Social Welfare, Mälardalen University, Universitetsplan 1, 72123 Västerås, Sweden; 4BUILD, Department of the Built Environment, The Faculty of Engineering and Science, Aalborg University, 2450 Copenhagen, Denmark; vll@build.aau.dk (V.L.L.); sdg@build.aau.dk (S.G.); 5Center for Clinical Research and Prevention, Copenhagen University Hospital, Bispebjerg and Frederiksberg, Hovedvejen, Indgang 5, 1. Sal, Nordre Fasanvej 57, 2000 Frederiksberg, Denmark; peter.bentsen@regionh.dk; 6Department of Geoscience and Natural Resource Management, University of Copenhagen, Rolighedsvej 23, 1958 Frederiksberg, Denmark; 7Necto, Denmark Gl. Kongevej 11-13, 1610 Copenhagen, Denmark; yun.ladegaard@necto.info; 8Department of Psychology, University of Copenhagen, 1353 Copenhagen, Denmark; 9Copenhagen Business Academy, Nansensgade 19, 1366 Copenhagen, Denmark; hhn@cphbusiness.dk; 10Research Centre for Social Sustainability, Department of People and Technology, Roskilde University, 4000 Roskilde, Denmark; katia@ruc.dk

**Keywords:** human nature interactions, nature connectedness, stress, sustainable work life, work environment

## Abstract

**Background/Objectives:** Outdoor office work (OOW) has been shown to promote health and well-being and to reduce stress. However, few empirical studies have examined research-based, simple approaches to implementing OOW. In preparation for a larger study, we conducted a feasibility study focusing on limited efficacy testing of potentially relevant outcomes for future OOW research. **Methods:** The simple *Pop Out* OOW programme consists of three workshops and access to online tutorials designed to support employees in transitioning relevant everyday office tasks outdoors. Before and after a 12-week intervention, employees from five small- and medium-sized Danish companies (*N* = 70) reported their weekly number of days including OOW, connectedness to nature (CNS and INS), Perceived Stress Scale (PSS), and well-being (WHO-5) scores. **Results:** At baseline, higher CNS scores were associated with a greater number of days including OOW per week (r = 0.25, *p* = 0.020). Following the intervention, participants reported a significant increase in the number of days per week with OOW (*p* < 0.01, d = 0.65). CNS scores also increased significantly (*p* = 0.019, d = 0.32). No significant changes were observed in stress or well-being scores across the entire sample. However, participants with PSS scores exceeding a national Danish criterion for high stress (*n* = 11) exhibited a significant and substantial reduction in perceived stress (*p* < 0.01, d = 1.00). **Conclusions:** Days including OOW, along with PSS and CNS scores, may serve as relevant outcome measures in future studies evaluating interventions aimed at promoting OOW. These outcomes should be assessed in larger and more diverse and controlled samples to establish generalisability.

## 1. Introduction

Work-related stress has become a pressing concern, with a growing number of employees reporting feelings of anxiety, anger, and worry in recent years, as demonstrated in the UNDP Human Development Report [1]. Mental health encompasses more than the absence of illness; it includes overall well-being, the ability to cope with life’s stresses, and meaningful participation in work [2]. The workplace significantly influences mental health through its organisation, culture, and leadership [2,3]. Establishing favourable workplace conditions is crucial for promoting both employee well-being and productivity [4,5,6,7]. A shift toward a salutogenic approach in work environment research has led to increased interest in nature-based interventions at work, which aim to enhance well-being and foster recovery [8,9,10]. As a specific subarea, research in outdoor office work (OOW) is accumulating promising findings [11,12,13,14,15,16,17]; contact with nature—meaning living elements or systems of plants and animals with a variety of cultural imprints, ranging from a rooftop terrace to an urban harbour to a woodland, as well as abiotic elements including the sky and celestial bodies—has, in general, been associated with numerous potential health benefits, including reduced levels of anxiety, stress, and depression, as well as improved cognitive performance, which may be particularly relevant in today’s cognitively demanding work environments [18,19,20]. Even small measures of time spent outdoors may have positive effects, although research in this area is sparse. In a survey including nearly 20,000 responders, White et al. found that spending a minimum of 120 min per week in nature is linked to significantly better self-reported health and well-being [21]. Both physical exposure to nature and a sense of psychological connectedness to nature may contribute to improved well-being [22]. Nature connectedness—the psychological sense of being part of the natural world—has increasingly been recognised as beneficial for mental health. In workplace contexts, higher nature connectedness has been linked to reduced stress, enhanced mood, and improved overall well-being [23]. This connection is thought to buffer against job-related stress by promoting restorative experiences and psychological resilience [24]. For employees, fostering nature connectedness (e.g., through access to green spaces, nature-based breaks, or biophilic office design) may mitigate the negative effects of high job demands and support mental health, potentially leading to lower burnout and greater engagement [25]. However, the effects of nature exposure seem to vary depending on context, timing, and individual differences, and are likely to be mediated through several interacting pathways, including stress reduction, improved air quality, and sometimes increased physical activity [18,19,26]. The mechanisms at work in nature exposure need further clarification, especially in relation to OOW [11]. Traditionally, nature in the workplace has been viewed primarily as a setting for breaks. More recently, however, interest has grown in relocating everyday office tasks outdoors, suggesting that integrating nature contact into work routines may improve employee well-being as well as productivity.

Kaplan’s early work highlighted the benefits of nearby nature in the workplace [27]. Relatedly, Hu et al. found that knowledge workers’ proximity to greenspace contributes to the overall health of employees by reducing stress, restoring attention, and promoting physical activity [28]. Even seemingly small measures, like a window view of greenery, can have a positive effect on employee well-being [29,30]. A cross-sectional study conducted in Norway found that greater indoor nature contact—such as plants and views of greenery—was significantly associated with reduced job stress, fewer subjective health complaints, and lower levels of sickness absence. Interestingly, these effects were partially mediated by perceived organisational support [31]. Subsequent research has demonstrated that even virtual reality nature environments can offer restorative benefits and reduce stress [32]. Studies investigating the impact of access to green outdoor spaces during workday breaks also show support for employee well-being by reducing stress and enhancing attentional capacity [33]. For instance, a pilot study examining the feasibility and effects of a daily 10 min outdoor work break resulted in significantly reduced perceived stress scores compared to indoor breaks [34]. Thus, an accessible green outdoor environment at the workplace can play an important role for employees’ health and well-being [35], particularly concerning work-related stress [34]. Furthermore, spending time outdoors can benefit work outcomes, such as increased attention and improved cognitive task performance [36,37], as well as engagement and creativity at work [38,39,40,41]. The present study was inspired by the Swedish interactive research project StickUt Malmö [15]. This study explored the benefits and challenges of relocating office work outdoors. Fifty-eight employees participated in OOW workshops, discussions, and practical exploration. The qualitative findings demonstrated that a variety of office tasks could be successfully performed outdoors—either while walking or seated—with reported benefits such as enhanced focus, well-being, stress relief, autonomy, and improved social interaction [15]. Key enablers included access to green spaces, supportive leadership, and organisational culture, while a common barrier was the feeling of guilt or illegitimacy associated with working outside. A constructivist grounded theory study by further proposed a conceptual OOW model showing the interrelatedness of categories such as enhanced well-being, improved performance, a sense of freedom, and a potentially deeper connection with oneself, others, and nature. It also emphasised the importance of continuous learning and challenging traditional norms surrounding work [42].

In line with this, recent studies have found that OOW can reduce stress [13] and enhance both productivity and well-being [11]. However, certain conditions must be met—both in the physical environment and in organisational attitudes—for these benefits to be realised. Perceptions among colleagues and managers can strongly influence the experience, with feelings of guilt potentially hindering participation or enjoyment [11,12,14,15]. In an exploratory study, Söderlund et al. reported positive experiences among office workers who temporarily relocated their work to an outdoor public space on a university campus [17]. Additionally, Herneoja et al. highlighted the potential of outdoor areas adjacent to office buildings, including roof terraces, where architectural solutions such as canopies and windbreaks can help create suitable microclimates for outdoor work [43].

In summary, research in the field of outdoor office work is primarily qualitative, with a notable lack of studies evaluating the effectiveness of research-based interventions designed to promote outdoor office work using quantitative outcome measures.

The purpose of this study is to assess the feasibility of selected outcome measures in the context of an outdoor office work (OOW) intervention. It serves as a preliminary step towards a larger randomised controlled trial (RCT), with the objective of determining whether the selected outcome measures are suitable for capturing the potential effects of OOW.

The central research question guiding this feasibility study is as follows: are the selected outcome measures appropriate for evaluating the effects of an OOW intervention?

Based on previous qualitative findings, our hypothesis is that the outcome measures—time spent on outdoor office work (OOW), stress, quality of life, and connectedness to nature—which were identified as relevant through qualitative research will also show measurable effects when assessed using quantitative methods.

## 2. Materials and Methods

### 2.1. Design

In the present feasibility study, we employed an observational pre–post design, without a control group, which administers questionnaires on OOW, nature connectedness, stress, and overall well-being (see Section 2.4) before and after a 12-week, partially online-based OOW intervention. Participants were employees from small- and medium-sized Danish workplaces (ranging from 10 to 210 employees per company).

The present study is part of the Pop Out project, conducted between March and June 2022. It was inspired by the Swedish StickUt Malmö project [44,45] and developed by an interdisciplinary research team aiming to support employees in adopting OOW. The Pop Out project also includes a qualitative component investigating the use of, and preferences for, outdoor settings [14].

### 2.2. The Intervention

We planned the intervention from January to March 2022 and held preparatory meetings with managers at the participating companies to ensure commitment, align expectations, and appoint local ambassadors (See Appendix A: TIDieR Checklist). To inspire and guide employees throughout the process, we produced seven short inspirational tutorial videos (5–15 min each) covering various perspectives and approaches to implementing outdoor office work. The content was developed by researchers from the fields of landscape architecture, office architecture, outdoor pedagogy, workplace mental health, management, and organisational studies. The tutorial topics included the following: (1) Experiences from Sweden; (2) Pop Out—Getting off to a Good Start Working Outdoors; (3) Why Is It Challenging to Go Outside? And Where Should We Go?; (4) To Persist—From Experience to Good Habits; (5) What Do I Gain from Working Outdoors?; (6) What Work Tasks Can I Take Outside with Me?; (7) Get Started Working Outdoors. Additionally, a researcher within landscape architecture conducted an environmental assessment of the natural surroundings of the participating companies. This assessment informed the development of tailored recommendations, inspirational materials, and practical guidelines designed to help organisations effectively utilise their immediate outdoor environments for office work. This information was intended as a starting point, as participants were encouraged to follow their own preferences in their selection of outdoor settings for OOW.

We initiated the intervention in mid-March 2022 by presenting the first tutorial. In early April, we conducted a 45 min kick-off session at each workplace to introduce the intervention’s background. To inspire participants to engage in OOW, we displayed a poster featuring photographs of different settings and routes in a communal area within each office. We also provided a list of potential office tasks that could be performed outdoors, encouraging reflection on the alignment between the task, environment, and necessary equipment (Figure 1).

Consistent with the recommendations of Wessels et al., we encouraged employees to select one or two tasks to be undertaken outdoors during the initial weeks and to explore the possibility of increasing their outdoor work activities throughout the intervention period [46]. We facilitated the planning and sharing of these initial task selections during the kick-off sessions to foster a sense of ownership and commitment.

We conducted approximately 20 min of each kick-off session indoors, with the remaining time spent outdoors. To support the adoption of OOW, we made the tutorial videos and practical guidance available to participants throughout the intervention period. In line with [12], we took different measures to make the intervention an integrated part of the ordinary organisational process: an example is the equipment of local ambassadors with adaptable email templates to effectively communicate relevant information to participating employees. At the conclusion of the intervention, we consolidated all materials and presented them on the ‘Smut Ud’ website [47].

Approximately one month after the kick-off sessions, we hosted a 45 min webinar for all participants. The webinar facilitated the sharing of activities and experiences across organisations, addressed challenges in altering work habits, and introduced tools for organising and conducting outdoor meetings.

Two months after the initial kick-off, we conducted a 45 min outdoor workshop at each of the five companies. These concluding sessions focused on gathering participants’ experiences and reinforcing elements such as motivation and commitment to encourage the continuation of outdoor office work beyond the intervention period.

We presented a programme theory for the Pop Out study describing the hypothesised relations between input, the intervention, and the outcomes (see Figure 2).

Intervention activities comprised three workshops: a kick-off session at each workplace, a collective webinar for all participants, and a concluding workshop at each site. Additionally, an employee at each workplace volunteered as an ambassador to customise and distribute project email templates containing tutorial links, coordinate OOW activities, and liaise with management.

Short-term outcomes anticipated from pre- to post-intervention included increased engagement in OOW, enhanced nature connectedness, improved well-being, and reduced stress levels. Mid- and long-term outcomes were not measured, but we hypothesised that sustained focus on OOW would maintain or enhance these benefits over time.

### 2.3. Recruitment and Participants

#### 2.3.1. Recruitment

We invited private companies, non-governmental organisations (NGOs), and labour unions to participate based on the following criteria: (1) employing between 10 and 250 staff members; (2) demonstrating explicit management support for project participation; and (3) possessing outdoor spaces suitable for office tasks, defined as areas incorporating natural elements such as vegetation, water, wildlife, stones, and sky, combined with walkable surfaces and, optionally, seating, table space, and shelter. Examples included gardens, courtyards, terraces, streetscape greenery, harbour areas, parks, beaches, and forests shown to be suitable for OOW [14,15]. The wide range of qualities in the outdoor space in each case gave the participants a good opportunity to find settings and routes that matched their preferences.

We prioritised voluntariness throughout all recruitment stages (criterion sampling), as higher levels of voluntary participation are positively associated with motivation and training transfer among employees [48]. Management support was an inclusion criterion, recognising that leadership is a critical factor in driving positive organisational change [49]. A perceived lack of legitimacy has also been identified as a significant barrier to OOW [11,15,16].

However, we acknowledged that during complex daily routines, there would be situations where leadership could not offer full support. Similarly, complete voluntariness might not apply in instances such as project kick-off meetings, where all enrolled participants were expected to attend. Our rationale was that genuine freedom of choice exists only when participants are aware of their options. Furthermore, voluntariness may be challenged when individuals must conform to group activities, such as departmental meetings.

To reach relevant workplaces, we utilised social media channels and directly approached organisations that expressed interest and had diverse natural surroundings, aiming to secure a varied dataset for our research.

Among the eligible companies, we included five organisations, one of which had two distinct locations—resulting in six workplaces in total. These comprised two service companies, two contractor companies, and one trade organisation. We invited all volunteering departments within these organisations to participate in the project.

#### 2.3.2. Participants

A total of 89 workers voluntarily took part in the study via their work site and provided baseline measurements. Of these, *N* = 70 workers (70/89 = 79%) provided voluntary, informed, written consent for their data to be used in scientific research. Hence, we included data from *N* = 70 in our baseline analyses. A total of *n* = 56 of these, representing 3–17 workers from each company, provided both baseline and post-intervention data (57/70 = 80% retest rate).

### 2.4. Measures and Data Collection

#### 2.4.1. Background Data

To characterise our sample, we measured demographics (age, gender, education) and work-related variables (working hrs. per week, seniority [years of experience at the work site], employment form [permanent, time-limited], work status [leader or not]). In line with our purpose of exploring relevant measures, we collected other background data, which are not reported here (such as questions concerning the degree of perceived influence on daily tasks), since they were unrelated to the main variables reported here.

#### 2.4.2. Limited Efficacy Testing

For the limited-efficacy testing [50], our main outcome was the number of days per week including OOW, based on a self-reported estimate across the past two weeks. In addition, we included four validated secondary outcomes. The connectedness to nature scale (CNS; we used only a sum of item 2 and item 4, although the full CNS has been professionally back-translated, adjusted, and approved by the original research group; [51,52]).

We used only two CNS items for several reasons. First, we aimed to limit the data collection burden on our participants and thereby increase the response rate. Second, we hypothesised that item 2 (“I think of the natural world as a community to which I belong”) and item 4 (“I often feel disconnected from nature”) were acceptable in typical Danish worksite culture and potentially sensitive to change during a low-intensity OOW intervention. Conversely, we hypothesised that other CNS items would require a more intensive nature-based intervention (such as a nature retreat) to be sensitive enough to detect change (e.g., item 6: “I often feel a kinship with animals and plants”; item 7: “I feel as though I belong to the Earth as equally as it belongs to me”).

The two items we chose were broad, secularly phrased statements of connectedness and disconnectedness with nature, respectively, and both showed adequate factor loadings in the original series of validation studies [52].

Second, we tested the Inclusion of Nature in Self (INS) [53], because the INS is also commonly used in the field of nature-based intervention research and distinguishes itself by presenting images of varying degrees of inclusion of nature in the self, rather than verbalised items.

Thirdly, since stress is often a problem at worksites which OOW is hypothesised to counter, we wished to test a measure of stress and therefore included the widely applied Perceived Stress Scale (PSS; [54]), where PSS scores > 17 in Denmark are recommended by the National Medicine and Health Authority as a marker of ‘high stress’.

Finally, since participation in nature-based programmes is often hypothesised to improve quality of life, we also included the World Health Organization’s Quality of Life-5 in this feasibility study [55,56].

All measures were completed online in Danish versions, 1–2 weeks prior to and after the Pop Out intervention, respectively.

We examined pre–post changes (alpha = *p* < 0.05, two-tailed) for each outcome with paired t-tests and Bonferroni–Holm correction for multiple tests. We report effect sizes with Cohen’s *d*, and to explore factors potentially related to OOW, we also conducted Bonferroni–Holm-corrected marginal correlation tests between our main outcome (days per week with OOW), our secondary outcomes (CNS, INS, PSS, and WHO-5 scores), and three background variables (age, gender, education).

## 3. Results

### 3.1. Participants

The 70 employees reported a mean (M) age of 43.01 years (Standard Deviation, SD = 12.01; range = 22–70). They identified as women (*n* = 43; 61%) or men (*n* = 27; 39%) and worked 36.8 hrs. per week on average (SD = 1.8), corresponding nearly to full time (37 h) in Denmark. The majority were permanently employed (*n* = 66; 94%), while *n* = 4 (6%) were in time-limited positions. A small group of *n* = 3 (4%) were leaders. Employees showed a large variation in seniority at their current work site, ranging from 0 to 27 years of experience (M = 4.01; SD = 5.98).

### 3.2. Outdoor Work

Before the intervention, the employees had about one day with OOW per week (*N* = 70; M = 0.96; SD = 1.41). Men reported significantly more days with OOW (M = 1.58 days.; SD = 1.78) than women (M = 0.58, SD = 0.96), *p* < 0.01, *d* = 0.73. In exploratory tests, we found this tendency in 4/5 companies (two service companies and two contractor companies), while the trade company showed no tendencies for gender differences on OOW. Connectedness to nature scale (CNS) scores were significantly related to the number of days including OOW (*r* = 0.25, *p* = 0.020), indicating that more days with OOW per week is related to a stronger perceived connectedness to nature already before the intervention. Age, education, stress (PSS-scores), and quality of life (WHO-5-scores) were unrelated to days with OOW, *p*s > 0.18, *r*s < 0.16.

Employees reported significantly more days including OOW after the intervention (*n* = 57, M = 1.84; SD = 1.48) than at baseline (*n* = 57, M = 0.95; SD = 1.41), amounting to a significant increase with a medium effect size, *t*(1,56) = 3.84, *p* < 0.01, *d* = 0.51 (Table 1). This indicates that workers used OOW about one more day per week after the Pop Out programme.

### 3.3. Connectedness to Nature

Across employees, CNS scores increased significantly during the intervention period, *t*(1,55) = 2.43, *p* = 0.019, *d* = 0.32. Interestingly, when we explored if this increase was dependent upon the active use of Pop Out intervention elements, we found that participants who had accessed the online tutorials at least once, increased in their CNS scores significantly (*n* = 28, *p* = 0.004, *d* = 0.60), while those who did not access the online tutorials on OOW showed no significant changes in CNS scores (*n* = 28), *p* > 0.5. CNS scores were unrelated to age and gender (*p*s > 0.37) but related to education (*r* = 0.27, *p* = 0.026). This indicates that more years of education are related to higher CNS scores. Moreover, CNS scores were unrelated to stress (PSS) scores and quality of life (WHO-5) scores, *p*s > 0.6. As mentioned, however, they were related to the number of days including OOW per week, *r* = 0.25, *p* = 0.020. The convergent validity of the CNS was supported, since the CNS scores were related to the Inclusion of Nature in Self (INS) scores at baseline, Spearman’s *rho* = 0.53, *p* < 0.0001. At the same time, we found nearly no changes in average INS scores from baseline (M = 4.6, *SD* = 1.29) to post-intervention (M = 4.6, *SD* = 1.32), *p* = 1.0.

### 3.4. Stress and Quality of Life

At baseline, the employees showed relatively low PSS scores (*M* = 12.29; SD = 6.40) and relatively high WHO-5 scores (M = 68.64; SD = 16.07), indicating generally low levels of stress and high levels of quality of life, respectively. We did not find any significant changes in PSS or WHO-5 scores across employees, *p*s > 0.2. However, a subgroup of *n* = 11 (17%) workers showed high levels of stress (PSS scores > 17). A post hoc test of these 11 workers showed a significant decrease in perceived stress, *t*(1,10) = 3.32, *p* < 0.01, amounting to a large estimated effect, but with a large confidence interval, *d* = 1.00, 95% CI [0.25–1.72]. Hence, workers with high levels of stress decreased in stress during the intervention.

## 4. Discussion

In the present study, we conducted limited efficacy testing of potentially relevant outcomes for larger OOW studies. We examined measures of the amount of OOW per week, connectedness to nature, inclusion in nature, stress, and quality of life. The Pop Out intervention that we examined comprised three workshops and online tutorials provided to train employees in transitioning relevant routine office tasks to OOW. In brief, employees reported a significantly increased amount of OOW and significantly increased connectedness to nature after the programme. We did not find significant changes in measures of stress and quality of life in the full sample, while 11 employees who scored above a criterion for high levels of stress at baseline demonstrated a significant decrease in stress.

Our main hypothesis was that the Pop Out intervention would result in more days including OOW. This was supported, since employees included OOW on about one more day per week after the programme, which was a significant increase, *p* < 0.01. This suggests a positive effect of using the relatively simple Pop Out intervention to promote OOW for employees in small- and medium-sized companies. A few more hours spent outdoors per week might represent a sufficient amount to contribute to positive change. For example, White et al. found that at least two hours per week in nature is associated with better mental health outcomes [21]. However, we did not measure the specific time spent outdoors. Our survey question asked about the number of days per week including OOW, and each OOW activity may thus have included both very short and, in principle, day-long activities. Still, our main finding supports the hypothesis that the intervention would result in more days including OOW. Thus, inquiring about the number of *days* including OOW therefore seems to be a feasible and sensitive measure of OOW in studies that do not include a highly technical set up. To capture hours spent outdoors, among other aspects of the outdoor work activities, more accurately, Petersson Troije et al. employed a custom-made mobile application to facilitate self-reporting on OOW experiences [15]. However, participant engagement with the app declined over the project period (of 15 months). As technology advances, future intervention studies could explore the use of mobile applications, or other technical devices —potentially with built-in GPS functionality—to enhance the accuracy and ease of tracking OOW activities [57].

We also found a significant increase in the perceived connectedness to nature (the sum of two CNS items) across employees. Interestingly, an exploratory test indicated that this effect was driven by a highly significant increase in the participants (*n* = 28) who accessed the online tutorials on OOW at least once (*p* = 0.004, *d* = 0.60), while participants who did not engage with the online tutorials (*n* = 28) did not show any substantial or significant change in CNS scores, *p* > 0.5, *d* = 0.13. These results align with qualitative findings proposing that OOW may enhance individuals’ perceived connectedness to nature [11,16]. In general, increased connectedness to nature may also increase with greater exposure to nature [58]. Our exploratory finding could indicate that the use of the online OOW tutorials increased the amount of OOW and potentially thereby employees’ perceived connectedness to nature, but we cannot rule out that participants who were more committed to OOW both used the online site more and increased their perceived connectedness to nature due to higher project engagement or expectancy effects. In this regard, baseline CNS scores were significantly and positively related to the number of days with OOW per week (*r* = 0.25, *p* = 0.020), which supports a relationship between the amount of OOW and nature connectedness prior to the Pop Out project. Workers who felt more connected to nature, also worked more outside, although the direction of causality cannot be determined. CNS scores were not related to perceived stress or quality of life scores, but to education length, where higher education was indicative of higher CNS scores. Other studies have found significant associations between nature connectedness and well-being [22,59], but this relationship was not observed in our study.

We found virtually no change on the Inclusion of Nature in Self (INS) scale, and INS scores were unrelated to OOW. The INS is a more qualitative and abstract measure using personal identification with images (rather than, e.g., verbal items), and it seeks to measure more in-depth perceptions of relations to nature (*inclusion of self* in nature) than the CNS. In summary, the INS seems less sensitive to changes during this brief intervention than the CNS and not as strongly related to OOW. Speculatively, changes in INS scores may require more intensive nature-based interventions. Alternatively, the INS may not have appealed to the participants in the same way as ordinary questionnaires, given its more abstract nature. An extended version of the INS has been developed [60] that offers items with simpler graphical designs than the original drawings applied in the present study. Questions about participants’ experience with the CNS, INS, and other questionnaires of nature connectedness could be integrated into qualitative interviews in future studies.

Curiously, we found that men worked more outdoors (1.58 days/week) than women (0.58 days/week) at baseline, demonstrating a significant gender difference, *p* < 0.01, *d* = 0.73. As seen, it corresponded to one more day including OOW per week for men. We did not find other gender differences. After the intervention, we found no gender differences in weekly days including OOW, meaning that women showed a descriptively larger increase in outdoor work than men, although this did not amount to a significant Time x Group interaction. These exploratory findings yield no firm conclusions about gender differences in outdoor work. However, the interconnections between gender, health and nature are complex [35,61], and gender as a factor for OOW may be further explored in future studies. For instance, Rosa et al. found that women reported a stronger connection to nature and a preference for outdoor environments [62], yet they were less likely than men to engage in nature-based recreation. This suggests that gender differences can influence participation in outdoor activities, underscoring the need to consider such variables when analysing data related to outdoor office work. Additionally, Di Fabio and Rosen examined individual differences in connectedness to nature, noting that personality traits and gender can play significant roles in how individuals relate to natural environments [63].

We did not observe significant changes in stress (PSS) or quality of life (WHO-5) scores across all employees. This may be attributed to the relatively low stress levels and high quality of life levels reported by participants at baseline, suggesting limited scope for measurable improvement. Notably, stress scores were in fact significantly reduced for a subgroup of 11 employees who showed high stress (PSS > 17) at baseline, *p* < 0.01, *d* = −1.00. Considering this, potential stress-reducing effects of OOW may be more prominent for workers with high stress who participate in OOW interventions. Our post hoc analysis has limited generalisability, and this finding should be investigated further in samples with higher degrees of perceived stress. However, previous studies have demonstrated associations between nature contact and nature connectedness with levels of stress and well-being. For instance, research indicates that a stronger connection to nature is linked to lower levels of stress and anxiety, as well as increased well-being [64]. Additionally, performing activities in natural environments has shown to positively influence both well-being and nature connectedness [65]. Other workplace studies have highlighted the importance of selecting and examining relevant subsamples to obtain meaningful results [66]. Potential enhancement of quality of life, as well as stress-reducing effects of interventions, such as the Pop Out, should therefore not be rejected. Future studies may focus more on initial stress levels and individual engagement with intervention elements as potential factors for stress-related changes. However, we should interpret the results with caution due to the small sample size, the generally low degrees of perceived stress, and thereby the relatively large statistical uncertainty and limited generalisability. Consequently, future studies should aim to use larger samples and consider more intensive interventions (e.g., with more activities promoting and facilitating OOW) if the goal is to reduce stress for all employees.

### 4.1. Strengths and Limitations

Strengths of this study include the involvement of five different companies across six workplaces, the design of a systematic short-term intervention, and the use of validated outcomes relevant to the field. However, several limitations should be noted. First, the absence of a control group means that observed changes cannot be clearly attributed to the intervention or the increase in OOW alone. Non-specific factors such as expectancy effects, seasonality, or regression towards the mean may also have influenced the outcomes. Second, OOW was measured solely before and after the intervention period and solely through self-reporting, which limits the precision of exposure assessment. Future studies may advance the field by combining self-reported measurements of stress or connectedness to nature with objective measurements of OOW through, e.g., GPS-based tracking tools.

Given the use of criterion sampling in the current study and the structured nature of participation, future research should explore more diverse and voluntary sampling strategies to mitigate potential biases. Specifically, the limited participant freedom and predefined inclusion criteria may have increased the risk of social desirability bias, as participants could feel compelled to report positive mental health outcomes aligned with perceived expectations. Future studies should consider employing randomised or self-selection sampling across a broader occupational spectrum to enhance generalisability and reduce bias. Additionally, incorporating measures to assess and control for social desirability bias (e.g., social desirability scales) would strengthen the validity of outcome evaluations in interventions involving outdoor office work.

Future studies should more systematically assess intervention compliance and the use of specific intervention components. The generalisability of the findings may be limited, as the participating small- and medium-sized companies may differ in needs and constraints compared to larger organisations or public institutions. While leadership support and voluntary participation are considered strengths of the current study, the results may not translate to contexts where such support is lacking, or where outdoor work is mandated rather than chosen.

Finally, the intervention took place in Denmark, and its applicability in other countries or cultural contexts may require adaptation. A general limitation was the intervention’s duration; 12 weeks during spring is a relatively short period. A longer intervention would provide deeper insight into daily practices across seasons and allow participants more time to develop routines, adapt, and establish lasting habits around outdoor office work.

### 4.2. Implications for Practice and Research

As suggested by Bowen et al., future steps may involve conducting additional small-scale experiments with diverse samples or subgroups [50], for example, through focusing more on initial stress levels and individual engagement with intervention elements as potential factors for stress-related changes or examining gender as a factor influencing OOW. To assess the intervention’s effectiveness across different contexts, the relationship between various types of OOW tasks and mental health outcomes could be investigated. This supports the notion that, regardless of the strength of the evidence underlying the identified principles, change practitioners must still adapt the guidance to the specific situations they encounter [67]. To evaluate the broader impact, large-scale randomised controlled trials (RCTs) involving multiple organisations and incorporating population-based surveys and data collection methods will be necessary [50]. Future studies should aim to use larger samples and may also consider more intensive interventions (e.g., incorporating additional activities that promote and facilitate OOW) with long-term follow-up if the goal is to reduce stress for all employees.

In future studies, data collection methods should more systematically assess intervention compliance and utilise alternative data collection techniques, such as mobile applications with GPS tracking and digital logging, to gather more precise and comprehensive data.

Qualitative data collection could also be included to investigate participants’ experiences with questionnaire items concerning nature relatedness—such as some of the more spiritual items from the connectedness to nature scale (CNS), which we hypothesised might be challenging in a Danish worksite context—as well as the visual images from the Inclusion of Nature in Self (INS) scale.

The aim of further qualitative research on these or similar scales could be to systematically identify items or scales that are most appropriate to the culture and intervention context. In the present study, we identified two items (items 2 and 4) on the CNS, which we hypothesised to be meaningful, acceptable, and sensitive to change during a low-intensity OOW intervention. The significant increase we observed in the sum score of these two CNS items supports this hypothesis; however, validation of these two items (or a brief version of the CNS) is also necessary to assess their validity.

In contrast, we found virtually no change on the INS, which does not support the sensitivity of the INS images to changes in participants’ relationship with nature following a brief OOW intervention.

Finally, we only examined measures of OOW, nature connectedness, stress, and quality of life. Evaluating additional parameters, such as measures of mental health and creativity, is also recommended, as nature-based interventions in workplace settings have been linked both to improved mental health as well as creativity [38,39,40,41,68].

## 5. Conclusions

In this observational, limited efficacy study, we investigated a series of potentially relevant self-reported outcomes to studies wishing to measure the effectiveness of low-intensity interventions aimed at supporting employees in using outdoor office work (OOW). We found change sensitivity on two outcomes: First, the number of self-reported days with OOW for the past two weeks increased significantly across all employees. Secondly, a simple sum score of two general, secular items on the connectedness to nature scale (CNS) increased significantly during the low-intensity Pop Out programme. On the contrary, we found no change on the Inclusion of Nature in Self (INS) scale, which may be explained both by lack of sensitivity of the INS and the low intensity of the Pop Out programme. We found no significant changes on the Perceived Stress Scale (PSS) or the WHO-5 Quality of Life index, but our sample showed averagely low stress scores and high quality of life already at baseline, perhaps creating floor/ceiling effects. In that regard, a post hoc analysis showed that participants with high stress scores reported significantly reduced stress. With the small sample size and the limited generalizability in mind, we conclude that the number of days including OOW, connectedness to nature (CNS), and perceived stress (PSS) are feasible and potentially sensitive outcome measures when investigating the effectiveness of a simple programme promoting OOW, such as Pop Out. However, larger and more controlled studies must be carried out to support such conclusions.

## Figures and Tables

**Figure 1 healthcare-13-01677-f001:**
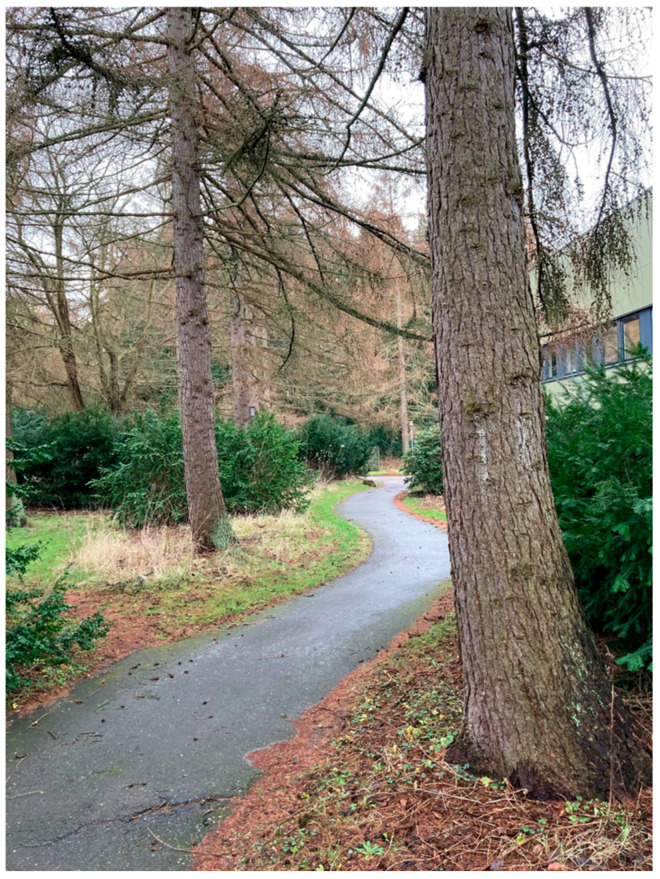
Example of aligning task, environment, and equipment: A route in the vicinity of one of the workplaces. Task: status meetings with colleagues; suitable environment: a park path allowing side-by-side walking; equipment: season-appropriate jacket and sturdy footwear.

**Figure 2 healthcare-13-01677-f002:**
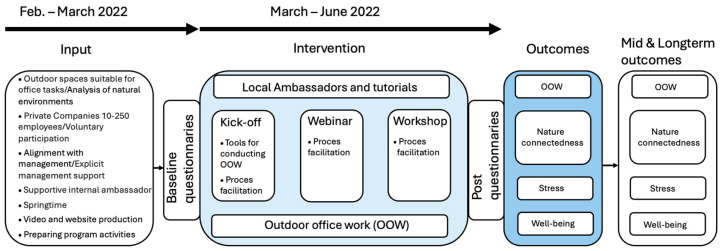
Programme theory of the Pop Out project. Inputs included the recruitment of six small- to medium-sized workplaces and site visits conducted by a landscape architecture researcher who assessed the natural environment for potential OOW opportunities. Our project facilitators met with management and designated workplace ‘ambassadors’ to establish commitment and define roles. We developed online tutorial videos and designed the programme to address barriers to OOW, aiming to inspire and engage employees.

**Table 1 healthcare-13-01677-t001:** Pre–post outcome changes in the Pop Out limited efficacy study of outdoor office work.

Days with Outdoor Office Work per Week	M	(SD)	*d* [95% CI]
*Baseline*	0.96	(1.41)	
*Post-intervention*	1.84	(1.50) **	0.51 [0.23–0.78]
Connectedness to Nature Scale (CNS)			
*Baseline*	7.36	(1.84)	
*Post-intervention*	8.09	(1.74) *	0.32 [−0.05–0.59]
Perceived Stress Scale (PSS)			
*Baseline*	12.16	(6.22)	
*Post-intervention*	11.50	(6.40)	0.12 [−0.10–0.35]
Quality of Life (WHO-5)			
*Baseline*	67.94	(15.16)	
*Post-intervention*	68.07	(16.01)	0.04 [−0.19–0.27]
Inclusion in Nature Scale (INS)			
*Baseline*	4.40	(1.29)	
*Post-intervention*	4.61	(1.32)	0 [−0.21–0.21]

Notes. Total baseline *N* = 70. Total post-treatment *n* = 56. * *p* < 0.05; ** *p* < 0.01. (Two-tailed, indicating significant change from baseline to post-intervention, Bonferroni–Holm-corrected).

## Data Availability

The data supporting the conclusions of this article will be made available from the corresponding author upon request.

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
