# Peer review of "Assessing an Outdoor Office Work Intervention: Exploring the Relevance of Measuring Frequency, Perceived Stress, Quality of Life and Connectedness to Nature"

_healthcare, 2025, doi:10.3390/healthcare13141677_

Round 1
Reviewer 1 Report
Comments and Suggestions for Authors
I have reviewed the manuscript titled “Introducing Outdoor Office Work: Testing Relevant Mental Health Outcome Measures.” The topic is timely and original, and the intervention is thoughtfully designed. The use of validated psychological scales and the focus on workplace well-being are clear strengths. However, there are important limitations that should be more clearly acknowledged. Most notably, the absence of a control group makes it difficult to attribute the observed changes directly to the intervention. Also, the sample largely includes participants with low baseline stress and high well-being, which may limit the relevance of the findings for those at greater mental health risk. All outcome data were self-reported, including the measure of outdoor office work days. In future studies, more objective methods like tracking tools or physiological markers would provide more robust evidence. Additionally, while nature connectedness is an interesting variable, its role in workplace mental health deserves more explanation. The manuscript would benefit from a clearer discussion of these limitations, and a stronger “future work” section. It’s also important that the authors clearly define the gap in the literature their study addresses. Suggesting next steps such as a randomized controlled trial with a more diverse population and longer follow-up would make this contribution even more valuable.
Comments on the Quality of English LanguageThe English could be improved to more clearly express the research.
Author Response
We sincerely thank Reviewer 1 for the valuable feedback and critical evaluation of our work.
The reviewer raised important points regarding control groups, self-reported vs. objective data and more, which we have carefully addressed in the revised manuscript. Detailed responses are provided in the attached file.

Reviewer 2 Report
Comments and Suggestions for Authors
Dear authors,
I read with great interest your mansucript titled "Introducing Outdoor Office Work: testing relevant mental health outcome measures". The feasibility study is well presented.
I have only some minor concerns that should be addressed:
- The title could be fine-tuned to represent the content of the study more specifically.
- The concept of OOW requires a clear definition. Is a roof-top terrace at a busy co-working space in the center of Barcelona a "suitable outdoor space for office tasks"? An even more suitable one than the park path you show in Fig. 1 because it has desk and chairs? Since you state that CNS scores were not related to perceived stress or quality of life scores, it seems that it's not relevant if it's green (forest, plants) or blue (lakes, sea, river, etc.) nature spaces, OR just being "outdoor". A clear definition is therefore necessary.
- "The Connectedness to Nature Scale [CNS; we used a sum of item 2 and item 4, [...]". Please justify clearly why.
- "Establishing favorable workplace conditions is crucial for promoting both employee well-being and productivity (Barling et al., 2004)." Pleasee provide a more recent reference, additionally.
- You mention yourself that criterion sampling and potential lack of "freedom" might bias the results, and I agree. After all, it seems that the sampling methodology does not prevent - rather increase - the risk of social desirability bias. Big limitation. Future research recommendations should be stated.
- It looks like you followed the steps of Stouten et al. (2018) rigorously (Stouten, J., Rousseau, D. M., & De Cremer, D. (2018). Successful Organizational Change: Integrating the Management Practice and Scholarly Literatures. Academy of Management Annals, 12, 752-788.
https://doi.org/10.5465/annals.2016.0095). Maybe you want to cite it or revise it.
Best of luck!
Author Response
We are grateful to Reviewer 2 for their insightful and helpful comments.
The reviewer’s input has led us to clarify several aspects of our methodology and discussion, and we believe the manuscript is significantly improved as a result. Our point-by-point responses to the reviewer’s suggestions are outlined in the attached file.

Reviewer 3 Report
Comments and Suggestions for Authors
The authors have addressed a particularly important topic for the business environment and the scientific community.
To improve the quality of the article, I recommend:
Line 142. Please explicitly describe the purpose of the research, the research questions, the objectives and the research hypotheses.
Reduce the generalizability of the results (especially in relation to stress).
Conclusions should capture all relevant aspects of the research, so written more broadly.
Study limitations. Were the psychological effects of COVID-19 still present during the experiment? Please discuss them, as well as non-specific factors (expectation effects, seasonality, etc.).
Limitations: The article could more explicitly emphasize the limitation regarding the lack of a control group in the conclusions and not only in the discussion section.
Limitations: Discuss whether it is worth maintaining the INS scale in future studies, given the lack of variation.
Author Response
We thank Reviewer 3 for recognising the importance of the topic and for the thoughtful and constructive feedback.
We appreciate your careful reading of our manuscript and your suggestions, which have helped us to improve the clarity and rigour of our work. In particular, your recommendations concerning the articulation of research aims, the interpretation of results, and the expansion of the limitations and conclusions sections were very helpful. We have addressed each of your points in detail below and revised the manuscript accordingly.

Round 2
Reviewer 1 Report
Comments and Suggestions for Authors
The authors have completely addressed all my comments, and I have no further concerns. Therefore, I recommend accepting the paper.
Comments on the Quality of English LanguageThe English could be improved to more clearly express the research.
Author Response
Dear Reviewer,
Thank you for your valuable comments during the process and for helping us improve the manuscript.
It has been a pleasure working with you.
On behalf on the author team,
Dorthe Djernis
Reviewer 3 Report
Comments and Suggestions for Authors
Congrats!
Author Response

(The authors gave the same response as above.)
